# Clinical and Molecular Spectrum of Sporadic Vascular Malformations: A Single-Center Study

**DOI:** 10.3390/biomedicines10061460

**Published:** 2022-06-20

**Authors:** Andrea Diociaiuti, Roberta Rotunno, Elisa Pisaneschi, Claudia Cesario, Claudia Carnevale, Angelo Giuseppe Condorelli, Massimo Rollo, Stefano Di Cecca, Concetta Quintarelli, Antonio Novelli, Giovanna Zambruno, May El Hachem

**Affiliations:** 1Dermatology Unit and Genodermatosis Unit, Genetics and Rare Diseases Research Division, Bambino Gesù Children’s Hospital, IRCCS, Piazza Sant’Onofrio 4, 00165 Rome, Italy; roberta.rotunno@opbg.net (R.R.); claudia.carnevale@opbg.net (C.C.); may.elhachem@opbg.net (M.E.H.); 2Translational Cytogenomics Unit, Multimodal Medicine Research Area, Bambino Gesù Children’s Hospital, IRCCS, Piazza Sant’Onofrio 4, 00165 Rome, Italy; elisa.pisaneschi@opbg.net (E.P.); claudia.cesario@opbg.net (C.C.); antonio.novelli@opbg.net (A.N.); 3Genodermatosis Unit, Genetics and Rare Diseases Research Division, Bambino Gesù Children’s Hospital, IRCCS, Piazza Sant’Onofrio 4, 00165 Rome, Italy; agiuseppe.condorelli@opbg.net (A.G.C.); giovanna.zambruno@opbg.net (G.Z.); 4Interventional Radiology Unit, Department of Imaging, Bambino Gesù Children’s Hospital, IRCCS, Piazza Sant’Onofrio 4, 00165 Rome, Italy; massimo.rollo@opbg.net; 5Department Onco-Haematology, Cell and Gene Therapy, Bambino Gesù Children’s Hospital, IRCCS, Piazza Sant’Onofrio 4, 00165 Rome, Italy; stefano.dicecca@opbg.net (S.D.C.); concetta.quintarelli@opbg.net (C.Q.); 6Department of Clinical Medicine and Surgery, University of Naples Federico II, Via Sergio Pansini 5, 80131 Naples, Italy

**Keywords:** diffuse capillary malformation with overgrowth, megalencephaly–capillary malformation–polymicrogyria syndrome, CLOVES syndrome, blue rubber bleb nevus syndrome, Parkes Weber syndrome, Klippel–Trenaunay syndrome, somatic mutation, *PIK3CA*-related overgrowth, *RASA1*, *TEK*, *KRAS*

## Abstract

Sporadic vascular malformations (VMs) are a large group of disorders of the blood and lymphatic vessels caused by somatic mutations in several genes—mainly regulating the RAS/MAPK/ERK and PI3K/AKT/mTOR pathways. We performed a cross-sectional study of 43 patients affected with sporadic VMs, who had received molecular diagnosis by high-depth targeted next-generation sequencing in our center. Clinical and imaging features were correlated with the sequence variants identified in lesional tissues. Six of nine patients with capillary malformation and overgrowth (CMO) carried the recurrent *GNAQ* somatic mutation p.Arg183Gln, while two had *PIK3CA* mutations. Unexpectedly, 8 of 11 cases of diffuse CM with overgrowth (DCMO) carried known *PIK3CA* mutations, and the remaining 3 had pathogenic *GNA11* variants. Recurrent *PIK3CA* mutations were identified in the patients with megalencephaly–CM–polymicrogyria (MCAP), CLOVES, and Klippel–Trenaunay syndrome. Interestingly, *PIK3CA* somatic mutations were associated with hand/foot anomalies not only in MCAP and CLOVES, but also in CMO and DCMO. Two patients with blue rubber bleb nevus syndrome carried double somatic *TEK* mutations, two of which were previously undescribed. In addition, a novel sporadic case of Parkes Weber syndrome (PWS) due to an *RASA1* mosaic pathogenic variant was described. Finally, a girl with a mild PWS and another diagnosed with CMO carried pathogenic *KRAS* somatic variants, showing the variability of phenotypic features associated with *KRAS* mutations. Overall, our findings expand the clinical and molecular spectrum of sporadic VMs, and show the relevance of genetic testing for accurate diagnosis and emerging targeted therapies.

## 1. Introduction

Vascular malformations (VMs) comprise a large, heterogeneous group of developmental disorders of the circulatory system, characterized by structural and functional alterations of the blood and lymphatic vessels. They are classified based on the main vascular structure affected (capillaries, veins, arteries, or lymphatics), and may be simple, combined, or associated with extravascular anomalies [1]. VMs encompass a wide range of manifestations, from localized, isolated skin lesions to highly disabling complex disorders [2,3]. They variably affect the skin, musculoskeletal system, gastrointestinal tract, central nervous system, eyes, upper airways, etc. VMs can be inherited, but most often occur sporadically, and are caused by somatic mutations. Sporadic VMs comprise capillary malformations (CMs)—which may be either isolated or syndromic, the latter being exemplified by Sturge–Weber syndrome (SWS)—and different lymphatic, venous, and combined malformations. An additional group of sporadic VMs is part of the *PIK3CA*-related overgrowth spectrum (PROS), which is characterized by congenital or early-onset segmental or focal overgrowth. VMs belonging to the PROS include severe and complex disorders, such as megalencephaly–capillary malformation–polymicrogyria syndrome (MCAP), congenital lipomatous overgrowth, vascular malformations, epidermal nevi, scoliosis/skeletal and spinal (CLOVES) syndrome, and Klippel–Trenaunay syndrome [2,4]. Over the past decade, the key molecular pathways and genetic mutations that drive the development of most VMs have been identified. The two main molecular pathways deregulated in sporadic VMs are RAS/mitogen-activated protein kinase (MAPK)/extracellular signal regulated kinase (ERK), and the phosphatidylinositol 3-kinase (PI3K)/protein kinase B (AKT)/mammalian target of rapamycin (mTOR) pathway. In addition, the G-protein-coupled receptor signaling molecules (GNAQ/GNA11/GNA14) are involved in the pathogenesis of isolated or syndromic CMs [5]. Interestingly, several genes belonging to these pathways are also implicated in cancer, and the same variants have been identified in tumors and sporadic VMs [2,4,6]. Increased availability of molecular testing has improved the diagnosis of VMs, and has led to the repurposing of anticancer drugs for the treatment of these complex disorders [2]. However, the majority of the molecular genetic studies have focused on severe phenotypes of the sporadic VM spectrum [7,8].

We performed a cross-sectional study of a well-characterized cohort of 43 patients affected by a wide range of sporadic VMs, including relatively benign though disabling forms, who had received molecular diagnosis by high-depth targeted NGS over the past two years. Our study focused on the correlation between clinical and imaging features and the sequence variants identified in the affected tissues.

## 2. Materials and Methods

### 2.1. Study Design

This was a single-center cross-sectional study correlating clinical and imaging features with genetic findings in patients affected with sporadic VMs who were followed-up in the multidisciplinary Center for Vascular Anomalies of Bambino Gesù Children’s Hospital.

Patients with a diagnosis of sporadic VM confirmed by genetic testing between June 2019 and September 2021 were recruited. Genetic testing was selectively performed in patients with a clinical diagnosis or suspicion of a syndromic VM or a segmental VM involving deep structures. Diagnoses were formulated according to the ISSVA classification [1]. Exclusion criteria were a diagnosis of a vascular tumor or vascular anomaly caused by an inherited germline mutation, or refusal to provide informed consent for study participation. The study was approved by the Institutional Ethical Committee (protocol number 1661_OPBG_2018) and conducted in accordance with the Declaration of Helsinki. All participants or their legal guardians provided written informed consent before entering the study.

### 2.2. Clinical Evaluation

A standardized case report form was used to collect information on patient demographics, history, clinical features, and imaging findings at the last follow-up visit.

### 2.3. Molecular Genetic Testing

Following informed consent, enrolled patients had received a molecular diagnosis on genomic DNA isolated from an affected tissue biopsy and a paired peripheral blood sample. In one patient (Table 1, n. 26), DNA was isolated from mucosal epithelial cells obtained via a buccal swab. The affected tissue was skin in all patients. In addition, DNA was isolated from adipose tissue in one case.

DNA isolation from lesional tissue was performed with Qiagen columns (QIAamp DNA mini kit; Qiagen, Hilden, Germany) according to the manufacturer’s instructions, and DNA isolation from blood was performed with QIAsymphony or QIAcube (Automated extraction, Qiagen). DNA concentrations were measured using NanoDrop (Thermo Scientific, Thermo Fisher, Waltham, MA, USA). Libraries for sequencing were generated using custom panel NGS targeted resequencing (NimbleGen SeqCap Target Enrichment, Roche Diagnostics, Basel, Switzerland) with UDI primers (KAPA Unique Dual-Indexed, Roche Diagnostics). The genes included in the custom panel are listed in Appendix A). Targeted sequencing was performed on the NGS sequencing platform NextSeq550 (Illumina, San Diego, CA, USA). Twenty-four samples per experiment were run to optimize the coverage, which was about 2000× per sample. Sequencing artefacts or contaminations were excluded through parallel tissue and blood analysis for each sample. Variant calling was performed using a DRAGEN Somatic Pipeline (v3.6.3, Illumina) specific for mosaicism analysis. Sequence variants were reported if the overall coverage of the variant site was ≥ 30 reads, ≥ 5 reads reported the variant, and the percentage of variant reads was ≥ 1%. Previously undescribed variants were confirmed by repeating targeted sequencing. In selected cases (Table 1, n. 36, 39, 40), sequence variants were also confirmed by multiplex droplet digital PCR (ddPCR) experiments, as detailed in Appendix A).

## 3. Results

### 3.1. Patient Population

Overall, 43 patients (19 males and 24 females) were included in the study. Their mean age was 9.1 + 7.4 years (range 0.8–42). All patients were Italian except for three, who were of Moroccan, Indian, and Ethiopian origin, respectively. The demographic, clinical, imaging, and molecular genetic features of our patient population are summarized in Table 1. CMs were present in 37 patients, including 1 non-syndromic CM, 3 SWS, 9 CMs with overgrowth (CMO), 11 diffuse CMs with overgrowth (DCMO), 6 MCAPs, 3 with CLOVES syndrome, 2 with KTS, and 2 combined VMs. One patient had a microcystic lymphatic malformation, and one had a venous–lymphatic malformation with overgrowth. Finally, the series included two patients with blue rubber bleb nevus (BRBN) syndrome and two with Parkes Weber syndrome (PWS). Using high-depth targeted NGS with a customized gene panel, pathogenic or likely pathogenic variants were identified in affected tissues from all recruited patients. Twenty-seven variants were identified in six genes, of which three were previously unreported.

### 3.2. Sturge–Weber Syndrome

The three cases of SWS (Table 1, n. 2–4) had a homogeneous CM involving the face, including areas derived from the frontonasal placode and, to variable extents, the body and limbs, as well as glaucoma (Figure 1A). Two patients also suffered from seizures, and the third suffered from recurrent headache. Brain magnetic resonance imaging (MRI) confirmed the presence of leptomeningeal angiomatosis in the three cases (Figure 1B). In all patients, the recurrent *GNAQ*-activating mutation p.Arg183Gln was identified with an allele frequency ranging from 2 to 5% in DNA from the affected skin biopsy, but not from peripheral blood (Table 1; Appendix A).

### 3.3. Capillary Malformation with Overgrowth

Nine patients (Table 1, n. 5–13) had CM limited to one anatomical region (one limb), with proportionate overgrowth of the same area. CM appeared as a combination of reticulate and homogeneously stained areas in three cases (Figure 2). Leg length discrepancy was observed in five of six patients with CM of the lower limb, and managed with orthotic measures in all cases but one, which required surgery. Two patients (n. 11, 13) presented proximal cutaneous syndactyly of toes 2–3, and one (n. 13) had sandal gap, i.e., a widened first web space of the foot (Figure 2E). Interestingly, one patient (n. 11) had hyperthermia of the thigh with visible vein ectasia in the same area (Figure 2B). All patients were in good general health and did not complain of pain or other symptoms related to their VM. No arteriovenous fistulas were detected by Doppler ultrasonography, including in patient n. 11, in whom MRI showed ectatic and tortuous superficial and deep veins in the affected limb. Molecular genetic testing identified known mutations in the *GNAQ* and *PIK3CA* genes in the affected skin biopsies from six and two cases, respectively (Table 1; Appendix A). Notably, patient n. 11 carried a known pathogenetic missense mutation affecting codon 12 of *KRAS* gene. The estimated variant allele frequency in DNA from tissue was similar in all patients, ranging from to 2 to 5%. The mutations were undetectable in genomic DNA from peripheral blood (Table 1).

### 3.4. Diffuse Capillary Malformation with Overgrowth

Patients with CM affecting multiple anatomical regions and proportionate soft tissue overgrowth, but lacking symptoms typical of other diseases of the PROS spectrum (e.g., MCAP, CLOVES, or KTS) or of SWS, were classified as DCMO [9]. All 11 patients affected with DCMO (Table 1, n. 14–24) had diffuse reticulate CM, with focal areas of homogeneous stain in 6 cases, including 3 cases with centrofacial and/or philtrum stain (Figure 3). Three patients had overgrowth of one limb, two presented with overgrowth of an upper and lower limb, three had different combinations of limb and trunk or face overgrowth, two had hemihypertrophy, and the last had macrodactyly only (Figure 3A,E). Four patients had leg length discrepancy managed with shoe inserts. Sandal gap and proximal toe syndactyly were present in five and four patients, respectively. All patients were otherwise healthy, except patient n. 24, who manifested psychomotor delay, which was considered to be related to the concomitant presence of a germline 15q11.2 microdeletion known to be associated with neurodevelopmental disorders [10]. NGS identified previously described somatic mutations in the *GNA11* and *PIK3CA* genes in three and eight patients, respectively (Table 1; Appendix A). The mutations were detected in affected tissues in all patients, with an allele frequency ranging from 3 to 26%. In addition, the same mutation was also detectable in peripheral blood in three cases, at an allelic frequency of 1% (Table 1, n. 14, 21, and 22).

### 3.5. Megalencephaly–Capillary Malformation–Polymicrogyria Syndrome

The six patients affected with MCAP presented macrocephaly and various combinations of trunk, limb, and face overgrowth (Table 1, n. 25–30) (Figure 4 and Figure 5). The CM was diffuse, partially reticulate, and also involved the face in all cases (Figure 4 and Figure 5), with philtrum stain in two cases. Five patients had proximal toe syndactyly, and four had sandal gap (Figure 4B and Figure 5E). Psychomotor delay was detected in four cases. Interestingly, patient n. 28 presented an epidermal nevus of the lower eyelid (Figure 5B), and the previously reported patient n. 30 had blaschkoid hypochromic macules on the thighs [11]. Neuroimaging confirmed megalencephaly or hemimegalencephaly in all cases (Figure 4D), thickened corpus callosum in four (Figure 4E), polymicrogyria in two (Figure 4D), Chiari malformation type I in two (Figure 4F), ventriculomegaly/hydrocephaly in two, and syringomyelia in one. Pathogenic gain-of-function *PIK3CA* mutations were identified in affected tissue in all cases, with an allele frequency ranging from 3 to 21%; and in peripheral blood in two cases (Table 1, n. 26 and 30; Appendix A).

### 3.6. Congenital Lipomatous Overgrowth, Vascular Malformations, Epidermal Nevi, Scoliosis/Skeletal and Spinal (CLOVES) Syndrome, Klippel–Trenaunay Syndrome, and Combined Vascular Malformations

Of the three patients with CLOVES, the first (Table 1, n. 31) presented the characteristic combined capillary–venous–lymphatic malformation on the flank, lower limb progressive overgrowth contrasting with upper limb and shoulder girdle hypotrophy (Figure 6A), various foot malformations (i.e., hexadactyly, macrodactyly, and widened web spaces) (Figure 6B), scoliosis, multiple lipomas (Figure 6A), urogenital malformations, and psychomotor delay. Moreover, the patient underwent surgery for hip dislocation secondary to abdominal lipomatous masses. MRI documented multiple trunk and left buttock lipomatous overgrowths (Figure 6C), together with terminal filum lipoma. In addition, it showed the extension of the combined vascular malformation to the pelvis and retroperitoneum. Patient n. 32 presented CMs on the trunk with prominent veins, together with multiple lipomas, lower limb overgrowth, and foot malformations (i.e., triangular shape, sandal gap, and proximal syndactyly). Back ultrasound confirmed the presence of hypertrophic lipomatous tissue of the dorsal and lumbar regions. MRI was not yet performed due to the patient’s young age, relatively mild clinical features, and absence of symptoms. Finally, the last patient (Table 1, n. 33) has been previously reported [12]. NGS molecular testing performed on DNA from lesional tissue confirmed the presence of somatic recurrent gain-of-function *PIK3CA* mutations at an allelic frequency ranging from 5 to 46.5% (Table 1; Appendix A).

Two pediatric patients (Table 1, n. 34, 35) presented with a painful combined capillary–venous–lymphatic malformation, associated with lower limb overgrowth and leg length discrepancy (Figure 7A). Due to the risk of thromboembolic complications, both patients received antiaggregant therapy. Patient n. 34 required surgical treatment of a congenital portosystemic shunt affecting liver function. Patient n. 35 underwent episodes of thromboses successfully managed with low-molecular-weight heparins. In both patients, imaging features confirmed the clinical diagnosis of KTS, showing the presence of a complex combined VM with a persistent embryonic vein (Figure 7B). NGS genetic testing identified known somatic activating mutations in *PIK3CA*, which were detected at an allelic frequency of 2.7 and 6%, respectively, in the DNA extracted from skin biopsies (Table 1; Appendix A).

Three patients had combined vascular malformations (Table 1, n. 36, 37, 39), which were localized (Figure 7C) and accompanied by overgrowth of the involved body region in patients n. 37 and 39. Previously described *PIK3CA* somatic activating mutations were identified in the affected tissues in these patients, but not in blood (Table 1; Appendix A).

### 3.7. Blue Rubber Bleb Nevus Syndrome

The two patients with BRBN syndrome (Table 1, n. 40, 41) presented typical clinical features, i.e., multiple cutaneous and subcutaneous bluish papules and nodules, which increased with age and also involved the palmoplantar surfaces (Figure 8A,B,D), a congenital plaque-like larger lesion (Figure 8C), and gastrointestinal venous malformations. Patient n. 40, an adult female, suffered from chronic anemia and recurrent gastrointestinal bleeding, which required multiple endoscopic laser treatments. In this patient, MRI documented cerebral, hepatic, and bone osteolytic vascular nodules. Both patients had double variants in the TEK gene; they carried the known missense somatic mutation affecting Tyr897, in combination with two somatic missense variants: c.2800T > C (p.Ser934Pro) (patient n. 40) and c.2743C > A (p.Arg915Ser) (patient n. 41) (Figure 9). In each patient, the two variants were detected at the same allele frequency, i.e., 1%, by NGS. In addition, in patient 41, the two variants (i.e., p.Tyr897Phe and p.Arg915Ser) were present on the same reads (see Appendix A). The variants p.Ser934Pro and p.Arg915Ser have not been reported, and are not annotated in the gnomAD database of human variations (see Appendix A). The p.Arg915Ser was considered to be likely pathogenic, because mutations at codon 915 resulting in different amino acid changes (p.Arg915Cys, p.Arg915Leu) have been previously identified in BRBN patients [13]. On the other hand, c.2800T > C leads to a p.Ser934Pro missense change localized in the kinase insert domain (KID) of the Tie2 protein [14,15,16,17] (Figure 9). The p.Ser934Pro change was found to be “Deleterious” by the sequence-homology-based tools PROVEAN and SIFT, and “Probably Damaging” (score = 1) by the structure-homology-based tool PolyPhen-2. In addition, the tool CUPSAT predicted that the substitution p.Ser934Pro determines unfavorable torsion angles, which reduce the thermal and chemical stability of the protein. Thus, various computational evidence supports a deleterious effect of the c.2800T > C variant, which was considered to be likely pathogenic.

### 3.8. Parkes Weber Syndrome

The two patients with PWS (Table 1, n. 42, 43) showed limb CM, with a pale halo in case n. 42, associated with overgrowth and increased skin warmth of the corresponding anatomic region (Figure 10A), as well as several small patches and macules scattered on the trunk and limbs (Figure 10D,E). They did not have any family history of VMs or skin diseases. Patient n. 42 complained of fatigue and local pain during febrile episodes. Patient n. 43 reported exercise-worsened leg pain. MRI showed humeral and subclavian vein ectasia in patient n. 42 (Figure 10B), and abdominal and lower limb ectatic veins in patient n. 43 (Figure 10F). Doppler ultrasound documented enlarged arteries with high-flow humeral vein arterialization in patient n. 42 (Figure 10C), and arteriolovenulous shunts with vein ectasia in patient n. 43. NGS genetic testing identified the frameshift variant c.1570_1571insTA (p.Cys525fs*19) in exon 11 of the *RASA1* gene in DNA from affected skin (but not blood) of patient n. 42. This variant has not been reported in the literature or databases, including gnomAD (see Appendix A). As it causes a premature stop codon, it was considered to be pathogenic, also in view of the patient’s consistent clinical and imaging features. Interestingly, molecular testing disclosed a known *KRAS* pathogenetic variant c.183A > T (p.Gln61His) in DNA from the skin biopsy (but not the blood) of patient n. 43.

## 4. Discussion

Our case series comprised 43 patients affected with a wide spectrum of sporadic VMs, ranging from relatively benign conditions such as non-syndromic CM or CMO and DCMO, to severe and rarer syndromes, such as SWS, MCAP, CLOVES, KTS, PWS, and BRBN syndrome. Thus, our patient population is illustrative of the clinical and molecular heterogeneity of sporadic VMs. Although the phenotypic and molecular genetic features of the majority of our patients were consistent with previous literature, we also observed several unusual findings, which highlight rarely reported or novel clinical and genetic characteristics of selected sporadic VMs.

The vast majority of our case series presented CMs mostly associated with other anomalies. Our molecular findings further confirm that the recurrent somatic *GNAQ* pathogenic variant c.548G > A (p.Arg183Gln) and other missense mutations affecting codon 183 are the major genetic determinants of SWS, isolated CM, and CMO [5,18,19,20]. In these disorders, p.Arg183Gln was mostly detected at a low allele frequency in DNA from lesional skin biopsies, consistent with previous findings [5,19,20]. It has been suggested that phenotypic variability in sporadic vascular anomalies relates to the timing and site of mutation during embryonic development, combined with the mutation’s functional consequences [4]. SWS would result from very early mutations in *GNAQ* in progenitor cells differentiating in brain, choroid, and skin endothelial cells, while non-syndromic CM may be due to a late somatic *GNAQ* mutation [5]. Notably, a known missense mutation in *KRAS* was identified in a 7-year-old girl (n. 11, Table 1) diagnosed with CMO based on clinical and imaging features. In addition to representing known oncogenic mutations, mosaic variants in KRAS have been described in multiple conditions, including epidermal nevus, nevus sebaceous and Schimmelpenning syndrome, encephalocraniocutaneous lipomatosis, and phacomatosis pigmentokeratotica [21]. Specific to sporadic VMs, missense mutations at codon 12 of the *KRAS* gene have been associated with arteriovenous malformations of the brain [22] and with cutaneous sporadic VMs—mostly high-flow lesions [23]. However, the same variants have also been detected in rare patients with low-flow capillary and/or venous malformations [23,24,25]. Similar to reported cases [23], the affected skin of our patient n. 11 was warm to palpation—a finding typically associated with high-flow VMs. However, repeated Doppler ultrasound did not detect arteriovenous malformations in this child. An additional feature common to our case and a few described patients with *KRAS*-mutated low-flow sporadic VMs is vein ectasia in correspondence with the CM [23]. Characterization of additional cases and long-term follow-up will allow us to better define the phenotype of this subset of patients, including possible late-onset of high-flow VM. Finally, known mutations in the *PIK3CA* gene were identified in two of our CMO patients—one also presenting sandal gap and proximal toe syndactyly. Interestingly, foot anomalies were not detected in any of the six patients affected with CMO due to *GNAQ* variants. Overall, the genetic heterogeneity of CMO highlights the relevance of also performing molecular genetic testing in view of possible future targeted therapies.

PIK3CA was mutated in the majority (8 out of 11) of our DCMO patients. DCMO was originally associated with GNA11 mutations [26] and, more recently, with *PIK3CA* mosaic variants [27]. Notably, six out of eight DCMO patients carrying PIK3CA mutations in our series presented hand/foot anomalies, comprising macrodactyly, syndactyly, sandal gap, and triangular foot, while none of the *GNA11*-mutated patients had such malformations. The association of hand/foot anomalies with *PIK3CA* variants has been already reported in two DCMO patients [27]. Indeed, hand/foot abnormalities are typical manifestations of the PIK3CA-related overgrowth spectrum [28]. Altogether, these findings suggest that hand/foot anomalies in DCMO—and likely in CMO—patients point to a *PIK3CA* mutation.

MCAP, CLOVES, KTS, and combined VM cases in our series carried somatic PIK3CA mutations detected at variable VAF in affected skin, consistent with previous literature findings, and with the hypothesis that the type of mutation and its timing and site during embryogenesis are major determinants of disease phenotype [7,8,28,29,30,31]. In one case (Table 1 n. 18) DNA was extracted from two different samples: adipose tissue from an overgrowth area, and lesional skin. In this patient, VAF was much higher in adipocytes (17%) than in whole skin (2%), in keeping with findings in CLOVES [8]. Four out of six MCAP patients had non-hotspot *PIK3CA* mutations (p.Glu726Lys, p.Glu81Lys, p.Arg115Pro) previously reported to be associated with the MCAP phenotype [7]. Asymmetric overgrowth was predominantly left-sided in MCAP (5/6 cases), CLOVES (2/3 cases), and KTS (2/2 cases), as previously described [28]. In addition to typical clinical manifestations, two of our MCAP patients presented unusual features: an epidermal nevus in patient n. 28, and hypochromic macules in a blaschkoid pattern in patient 30 [11]. To our knowledge, epidermal nevus has not been described in MCAP, while it is a common feature of CLOVES [4]. In addition, isolated epidermal nevus can carry *PIK3CA* mutations [32]. Although further reports are needed to confirm the association of epidermal nevus and MCAP, our observation underlines the clinical overlap between different disorders belonging to the PROS spectrum.

Both patients with BRBN syndrome presented the typical phenotypic features and carried double somatic variants in the *TEK* gene. They had a recurrent mutation affecting Tyr897 [13], associated with the previously undescribed missense variants p.Ser934Pro in patient n. 40 and p.Arg915Ser in n. 41. *TEK* double mutations in cis are known to underlie BRBN syndrome [13]. In our patients, base changes occurred at the same frequency and, for patient n. 41, on the same reads, indicating that the identified somatic mutations were on the same allele. Moreover, the identification of two novel somatic sequence variants contributes to expanding the genotypic spectrum of BRBN syndrome.

Our patient n. 42 presented with the typical clinical and imaging features of PWS, in the absence of a family history of skin lesions and vascular diseases. PWS can occur as an isolated disorder or be part of a capillary–arteriovenous malformation (CM-AVM) [33]. CM-AVM is an autosomal-dominant disorder due to germline truncating mutations in *RASA1* or, less frequently, in *EPHB4* [2]. The presence of multifocal skin lesions in familial CM-AVM suggests that a two-hit mechanism underlies disease development, with a somatic mutation occurring at lesion sites in addition to the germline variant. Indeed, this mechanism has been demonstrated in a few *RASA1*-mutated patients [33,34,35]. In our sporadic patient with PWS, we identified a novel pathogenetic somatic frameshift *RASA1* variant, which was detected in lesional tissue, but not in blood. To date, a single case of sporadic PWS associated with a truncating mosaic *RASA1* mutation has been reported in the literature [36]. Moreover, seven patients with sporadic CM-AVM have been shown to carry a RASA1 mosaic variant in lesional tissue [35,36,37] and, in one case, an additional variant both in the blood and in the malformation [35], suggesting that the two-hit hypothesis may also hold valid for sporadic cases of PWS and CM-AVM. Therefore, we cannot exclude the possibility that we have missed a low-level second mosaic mutation in our patient. Finally, the recent demonstration of somatic mosaicism in the germline in a sporadic CM-AVM patient with three affected children has major implications for genetic counselling of sporadic cases of PWS and CM-AVM with regard to their reproductive risk [35].

Our last patient presented skin CMs and multiple arteriolovenulous shunts of the lower limb associated with ipsilateral limb overgrowth. These features are suggestive of mild PWS. Unexpectedly, a known mosaic *KRAS* variant was identified in lesional skin. Our patient’s phenotype is similar to that of a recently published child who also carried a *KRAS* somatic mutation [38]. Moreover, *KRAS* somatic mutations have been reported in two adult patients diagnosed with PWS who also presented unusually severe lymphedema and microcystic lymphatic anomalies [39]. Altogether, these findings further illustrate the variability and partial overlap of VM phenotypes associated with *KRAS* mutations.

The major limitation of our study was the relatively small number of cases affected by each type of sporadic VM. However, our patient cohort was clinically, radiologically, and genetically characterized by a multidisciplinary team in our Reference Centre for Rare Vascular Anomalies, ensuring accurate genotype–phenotype correlations. The majority of our patients were pediatric; thus, the reported clinical features are not representative of all of the complications that can occur in the progressive course of several sporadic VMs, such as those belonging to PROS, BRBNS, and PWS.

In conclusion, our case series contributes to expanding the clinical and genetic spectrum of sporadic VMs by the identification of three previously undescribed sequence variants in the *TEK* and *RASA1*, genes and by showing novel phenotypic features associated with *KRAS* somatic mutations. Moreover, we describe in detail the milder phenotypes associated with *PIK3CA* somatic mutations in CMO and DCMO patients, and the overlapping features between clinical subgroups. Overall, our findings further highlight how genetic testing is a key element of the diagnostic management for patients with sporadic VMs, as it allows a precise diagnosis, and represents the basis for emerging targeted therapies.

## Figures and Tables

**Figure 1 biomedicines-10-01460-f001:**
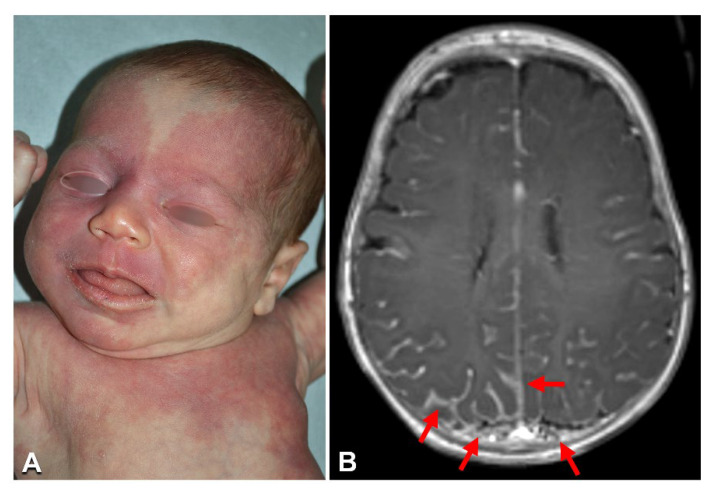
Clinical imaging of Sturge–Weber syndrome: A 2-month-old female (Table 1, n. 2) with a diffuse capillary malformation of the trunk, limbs, and face, where the upper eyelids and forehead are involved (**A**); brain MRI documenting the leptomeningeal angiomatosis (arrows in **B**) (**B**).

**Figure 2 biomedicines-10-01460-f002:**
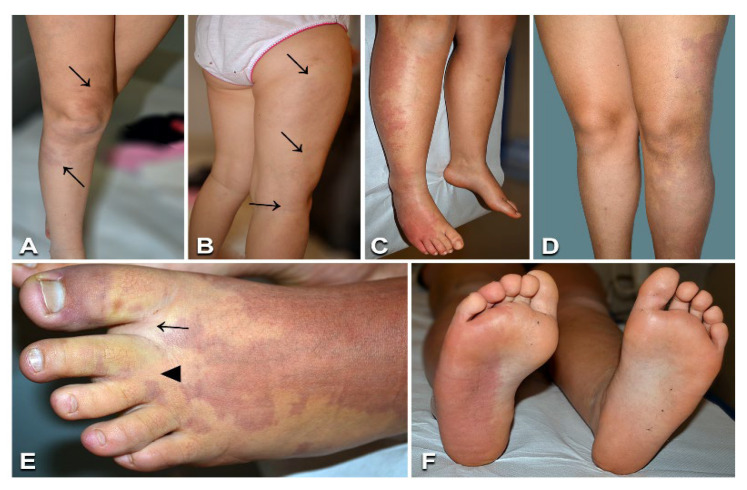
Clinical features of capillary malformations with overgrowth: A 7-year-old female (Table 1, n. 11) presenting red-brownish patches with blurred margins on the right thigh and leg (arrows in **A**), prominent veins on the lateral aspect of the thigh (arrows in **B**), and ipsilateral limb overgrowth. Eleven-year-old male (Table 1, n. 8) with a capillary malformation of the lateral region of the right lower limb, including the foot dorsum and plant (**C**,**F**), associated with ipsilateral lower limb overgrowth (**C**). Seventeen-year-old female (Table 1, n. 13) presenting a capillary malformation of the left lower limb, including the foot (**D**,**E**). Note sandal gap (arrow in **E**) and proximal syndactyly of toes 2–3 (arrowhead in **E**).

**Figure 3 biomedicines-10-01460-f003:**
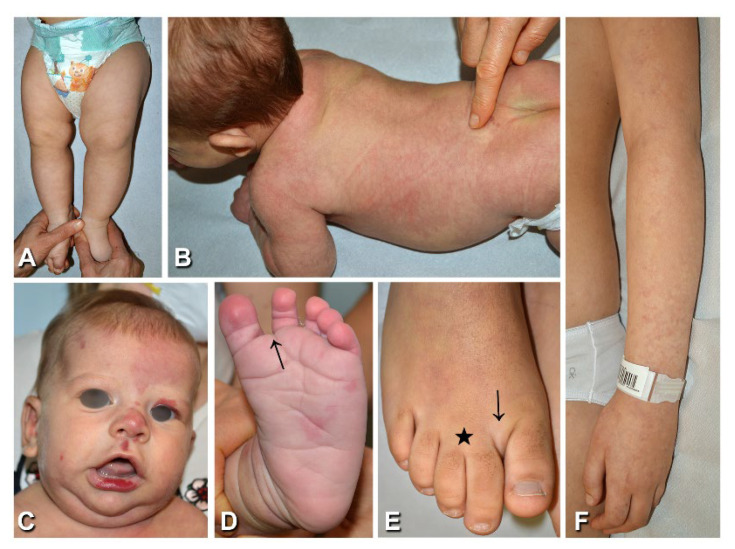
Clinical findings of patients presenting diffuse capillary malformation with overgrowth: Reticulate capillary malformation with overgrowth of the left lower limb in case n. 14 (Table 1), at 7 months of age (**A**); in this patient the capillary malformation was also present on the trunk and face, with vermillion stain. A 2-month-old patient (Table 1, n. 16) with a diffuse reticulate capillary malformation of the trunk and limbs (**B**), also involving the feet (**D**), and a homogeneous capillary malformation of the left forehead and centrofacial region, including philtrum and vermillion stain (**C**); note the sandal gap (**D,** arrow) and triangular shape of the left foot (**D**). Sandal gap (arrow), proximal syndactyly of toes 2–3 (asterisk), and macrodactyly of toe 2 (**E**) in a 10-year-old female (Table 1, n. 23). Reticulate capillary malformation of the left upper limb (**F**) in a 10-year-old female, who presented similar lesions over most of the body’s surface (Table 1, n. 22).

**Figure 4 biomedicines-10-01460-f004:**
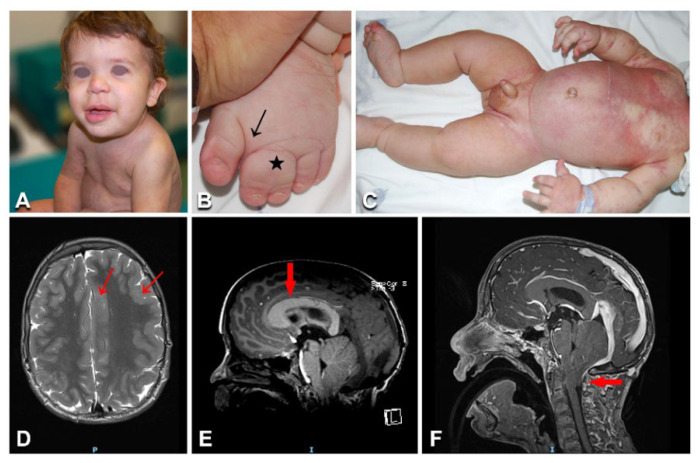
Clinical and imaging features in a toddler with megalencephaly–capillary malformation–polymicrogyria syndrome: (**A**) Diffuse partly reticulate capillary malformation on the trunk, limbs, and nose in patient n. 25 at the age of 17 months (**A**,**C**); the toddler also presents macrocephaly (**A**), along with left face and upper and lower limb hypertrophy (**A**,**C**), as well as sandal gap and syndactyly of left toes 2-3 (arrow and asterisk in **B**). Brain MRI shows left hemimegalencephaly (**D**), polymicrogyria (arrows in **D**), thick corpus callosum (arrow in **E**), and Chiari malformation type I with herniation of the cerebellar tonsils through the foramen magnum (arrow in **F**).

**Figure 5 biomedicines-10-01460-f005:**
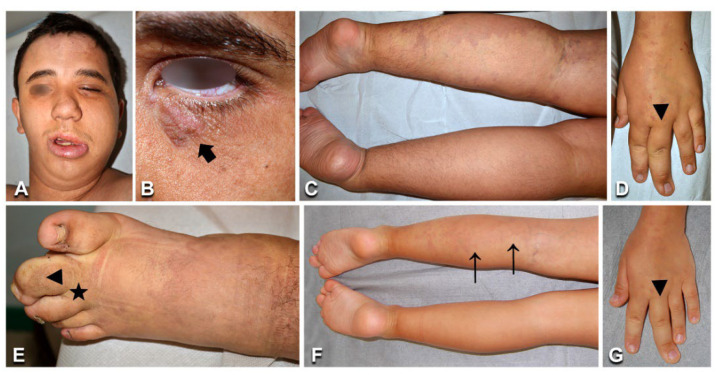
Clinical and imaging features in a young patient with megalencephaly–capillary malformation–polymicrogyria syndrome. Patient n. 28 presents a diffuse, partly reticulate capillary malformation (**A**,**C–E**, arrows in **F**,**G**), macrocephaly and left face hypertrophy (**A**), and an epidermal nevus on the left lower eyelid (**B**, arrow). Hand and foot anomalies include left third finger macrodactyly (arrowhead in **D**,**G**), left second toe macrodactyly, and proximal syndactyly of toes 2–3 (arrowhead and asterisk in **E**). The progressive overgrowth is documented in panels (**F**,**G**) versus panels (**C**,**D**), showing pictures taken at 6 and 19 years, respectively.

**Figure 6 biomedicines-10-01460-f006:**
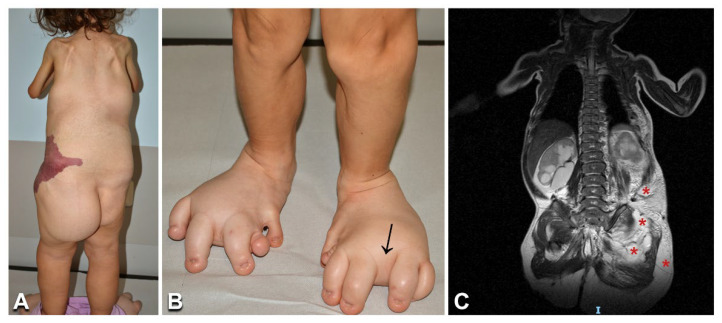
Clinical features and imaging findings in a 6-year-old girl with CLOVES syndrome: The child (Table 1, n. 31) presents the characteristic combined capillary–lymphatic–venous malformation on the left flank (**A**), multiple lipomas on the back and left buttock (**A**), and severe foot anomalies, including left hexadactyly, toe macrodactyly, and widened web spaces (arrow in **B**). Moreover, note the upper limb and shoulder girdle hypotrophy (**A**). Total body MRI shows multiple trunk and left buttock lipomatous masses (asterisks in **C**).

**Figure 7 biomedicines-10-01460-f007:**
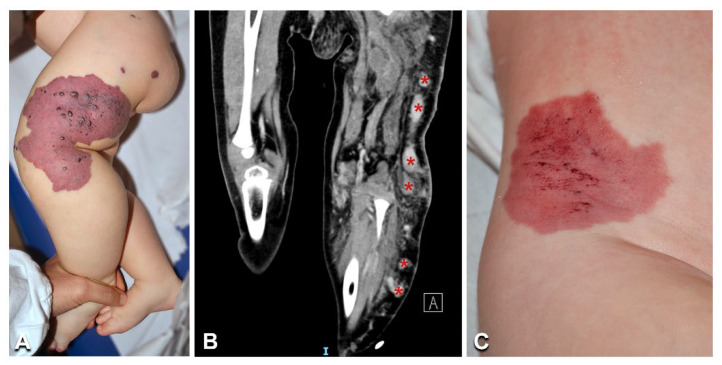
Clinical and imaging features of Klippel–Trenaunay syndrome: A 3-year-old girl (Table 1, n. 35) presents a combined capillary–venous–lymphatic malformation on her left buttock and lower limb (including the toes), with overgrowth (**A**). CT scan shows a persistent embryonic superficial vein (**B**, asterisks). Note the clinical similarity of the vascular malformation in Klippel–Trenaunay (**A**) and CLOVES (Figure 6A) with an isolated combined capillary–lymphatic malformation of the right flank in a 1-year-old male infant (Table 1, n. 36) (**C**).

**Figure 8 biomedicines-10-01460-f008:**
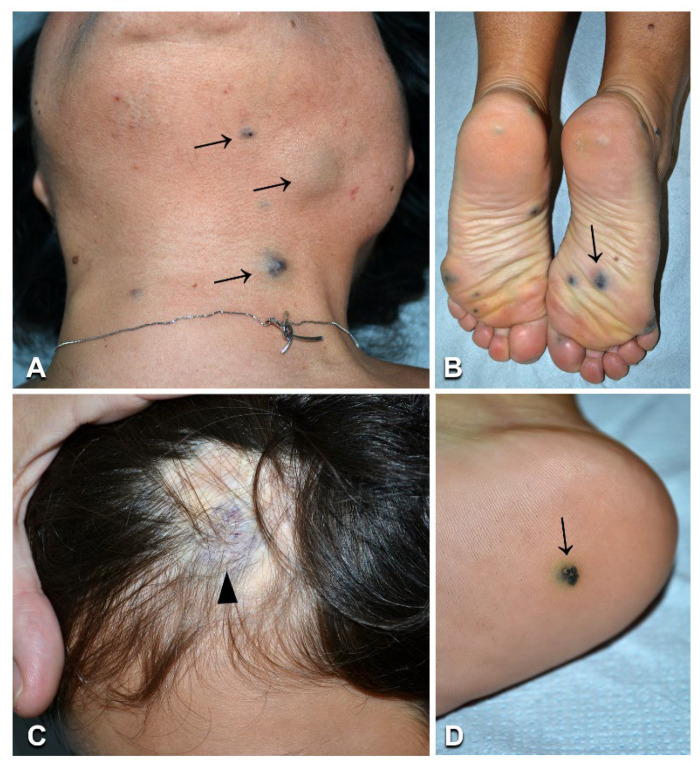
Clinical features of blue rubber bleb nevus syndrome: A woman aged 42 years (Table 1, n. 40) presents multiple cutaneous and subcutaneous bluish papules and nodules, including on plantar surfaces (arrows in **A**,**B**). A similar lesion is visible in a 9-year-old boy (Table 1, n. 41) (**D**, arrow), together with the characteristic violaceous congenital plaque on the left occipital area (**C**, arrowhead).

**Figure 9 biomedicines-10-01460-f009:**
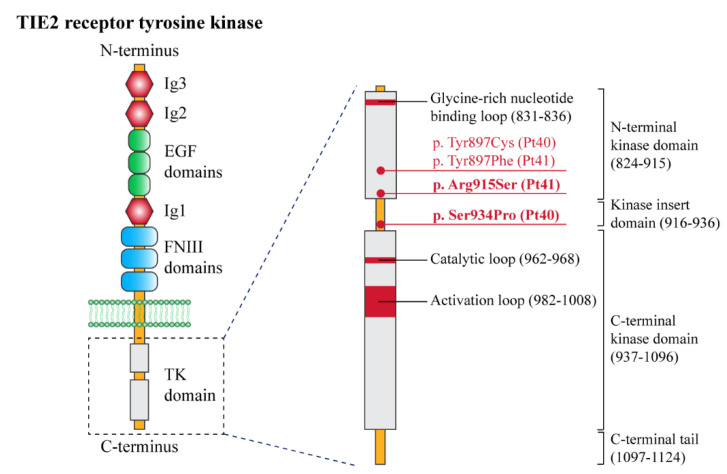
Structural organization of the receptor tyrosine kinase TIE-2 with the position of the pathogenic variants identified in this study. Left panel: schematic of the domain structure of the TIE-2 receptor. The extracellular portion of the receptor contains three epidermal growth factor (EGF)-like domains (green circles) flanked by three immunoglobulin (Ig)-like domains (Ig1, Ig2, and Ig3) (red hexagons), followed by three fibronectin type III (FNIII) domains. A single-pass transmembrane domain (phospholipidic bilayer is represented in green) precedes an interrupted tyrosine kinase (TK) domain (light grey) within the intracellular C-terminal tail. Right panel: detailed representation of the TK domain, with its functional sites, and the pathogenetic variants identified in this study (red dots). The new variants (p.Arg915Ser and p.Ser934Pro) are indicated in bold. TIE-2 domain amino acid numbers (bracketed) are based on literature data [14,17] and the UniProt database.

**Figure 10 biomedicines-10-01460-f010:**
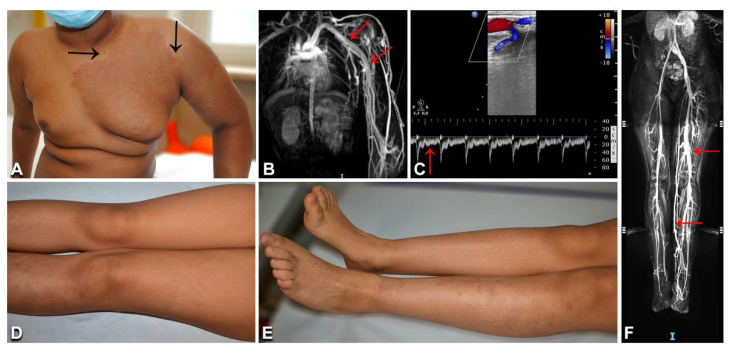
Clinical and imaging characteristics of Parkes Weber syndrome: Left upper limb and thorax reddish patch with pale halo (**A**, arrows), and hypertrophy of the affected area (**A**), in an 11-year-old Indian boy (Table 1, n. 42). In the same patient, thoracic and upper limb MRI shows humeral and subclavian vein ectasia (arrows in **B**); Doppler ultrasound documents an arteriovenous shunt and vein ectasia (**C**), as well as arterial high diastolic flow (arrow in **C**). A 12-year-old Moroccan girl (Table 1, n. 43) presents multiple brownish patches on the left lower limb, with ipsilateral overgrowth (**D**,**E**). Abdominal and lower limb MRI documents ectatic veins (**F**, arrows).

**Table 1 biomedicines-10-01460-t001:** Demographic, clinical, imaging, and molecular genetic features of patients with somatic vascular malformations.

Pt. n.	Age (y)/Sex	Diagnosis	Clinical Features	Imaging Findings	Gene	Variant(s) *	VAF (Tissue **/ Blood)
**1**	8/M	CM	Right lower face up to lower eyelid CM	Brain MRI: normal	*GNAQ*	c.548G > A (p.Arg183Gln)	17%/nd
**2**	0.8/F	SWS	Bilateral face, trunk, and limb CM; right eye glaucoma; seizures	Brain CT: subcortical calcifications; brain MRI: leptomeningeal angiomatosis	*GNAQ*	c.548G > A (p.Arg183Gln)	3%/nd
**3**	8/F	SWS	Left face and upper limb CM; left thorax and scapular prominent veins; left upper limb and thorax hypertrophy; left eye glaucoma; headache; epistaxis	Brain MRI: left cerebral hypotrophy; leptomeningeal angiomatosis	*GNAQ*	c.548G > A (p.Arg183Gln)	2%/nd
**4**	21/M	SWS	Right facial and hemibody CM; ipsilateral lip, upper and lower limb hypertrophy; right eye glaucoma; seizures	Brain MRI: leptomeningeal angiomatosis	*GNAQ*	c.548G > A (p.Arg183Gln)	5%/nd
**5**	4/F	CMO	Left chest, shoulder, and upper limb partly reticulate CM; ipsilateral limb overgrowth	DU: no AVF	*GNAQ*	c.548G > A (p.Arg183Gln)	3%/nd
**6**	15/M	CMO	Left lower limb CM and overgrowth; leg length discrepancy	DU: no AVF	*GNAQ*	c.548G > A (p.Arg183Gln)	3%/nd
**7**	12/F	CMO	Right chest, shoulder, and upper limb partly reticulate CM; ipsilateral limb overgrowth	DU: no AVF	*GNAQ*	c.548G > A (p.Arg183Gln)	5%/nd
**8**	11/M	CMO	Right lower limb CM and overgrowth; leg length discrepancy	DU: no AVF	*GNAQ*	c.548G > A (p.Arg183Gln)	2%/nd
**9**	14/F	CMO	Left lumbar and lower limb CM; ipsilateral limb overgrowth; leg length discrepancy (surgically treated)	DU: no AVF	*GNAQ*	c.548G > A (p.Arg183Gln)	5%/nd
**10**	14/F	CMO	Right chest and upper limb partly reticulate CM; ipsilateral limb overgrowth	DU: no AVF	*GNAQ*	c.548G > A (p.Arg183Gln)	3%/nd
**11**	7/F	CMO	Right lower limb CM and prominent veins, warmth to palpation; ipsilateral limb overgrowth; leg length discrepancy; proximal toe syndactyly	DU: no detectable AVF; MRI: right lower limb ectatic, tortuous superficial and intramuscular veins	*KRAS*	c.35G > T (p.Gly12Val)	3%/nd
**12**	4/F	CMO	Left lower limb CM and overgrowth	DU: no AVF	*PIK3CA*	c.1636C > A (p.Gln546Lys)	3%/nd
**13**	17/F	CMO	Left lower limb CM and overgrowth; sandal gap; proximal toe syndactyly; leg length discrepancy	DU: no AVF	*PIK3CA*	c.353G > A (p.Gly118Asp)	3%/nd
**14**	1/F	DCMO	Lower limb, trunk, and face reticulate CM with vermillion stain and prominent left face veins; left lower limb overgrowth	DU: no AVF	*PIK3CA*	c.1090G > A (p.Gly364Arg)	7%/1%
**15**	11/M	DCMO	Limb, trunk, and head reticulate CM; right finger and toe macrodactyly	DU: no AVF	*PIK3CA*	c.1133G > A (p.Cys378Tyr)	13.5%/nd
**16**	1/M	DCMO	Limb and trunk reticulate CM with centrofacial stain ***; left upper limb overgrowth; sandal gap; triangular foot	DU: no AVF; brain MRI: normal	*PIK3CA*	c.1093G > A (p.Glu365Lys)	7%/nd
**17**	6/F	DCMO	Trunk, left thigh, and cheek reticulate CM; abdominal wall prominent veins; overgrowth of left cheek, palatine tonsil, parotid gland, and paraumbilical and lumbosacral regions; sandal gap; proximal toe syndactyly	DU: no AVF; Brain MRI: temporal and occipital lobes asymmetry with left predominance	*PIK3CA*	c.1357G > A (p.Glu453Lys)	7%/nd
**18**	10/F	DCMO	Right hemibody reticulate CM; right lower limb prominent veins; upper limb and focal trunk overgrowth with toe macrodactyly; sandal gap; proximal toe syndactyly	DU: no AVF; brain MRI: normal	*PIK3CA*	c.1357G > A (p.Glu453Lys)	2%/nd
**19**	1/F	DCMO	Trunk and lower limb reticulate CM with centrofacial stain ***; right lower limb overgrowth	DU: no AVF	*PIK3CA*	c.2740G > A (p.Gly914Arg)	26%/nd
**20**	19/F	DCMO	Right hemibody partly reticulate CM; right upper and lower limb overgrowth; leg-length discrepancy	DU: no AVF; MRI and CT angiography: left internal carotid artery hypoplasia	*GNA11*	c.547C > T (p.Arg183Cys)	3%/nd
**21**	11/M	DCMO	Limb, trunk, and face reticulate CM; left upper limb and right lower limb overgrowth; leg length discrepancy	DU: no AVF; brain MRI: normal	*GNA11*	c.547C > T (p.Arg183Cys)	20%/1%
**22**	10/F	DCMO	Limb, trunk, and face reticulate CM; right lower limb and left cheek overgrowth; leg length discrepancy	DU: no AVF	*GNA11*	c.548G > A (p.Arg183His)	3%/1%
**23**	10/F	DCMO	Limb, trunk, and face reticulate CM; left hemihypertrophy and macrodactyly; sandal gap; proximal toe syndactyly	DU: no AVF	*PIK3CA*	c.1133G > A (p.Cys378Tyr)	11.5%/nd
**24**	6/M	DCMO	Face, trunk, and limb partly reticulate CM; right hemihypertrophy with macrodactyly; leg length discrepancy; proximal toe syndactyly; sandal gap; psychomotor delay ****	DU: no AVF; brain MRI: thick corpus callosum	*PIK3CA*	c.311C > T (p.Pro104Leu)	6%/nd
**25**	14/M	MCAP	Face, trunk, and limb partly reticulate CM; macrocephaly; left face, upper and lower limb, and right fingers hypertrophy; bilateral toe syndactyly; psychomotor delay	Brain MRI: left hemimegalencephaly; polymicrogyria; thick corpus callosum; Chiari malformation type I and obstructive ventricular dilation; cerebellar vein ectasia	*PIK3CA*	c.353G > A (p.Gly118Asp)	14%/nd
**26**	5/F	MCAP	Left hemibody partly reticulate CM; macrocephaly; overgrowth of left face, teeth, lower limb, and toes bilaterally; leg length discrepancy; sandal gap	Brain MRI: left hemimegalencephaly; thick corpus callosum	*PIK3CA*	c.2176G > A (p.Glu726Lys)	19%/4%
**27**	11/M	MCAP	Trunk, limb, and face partly reticulate CM; face and left lower limb prominent veins; macrocephaly; lower limb hypertrophy; upper limb hypotrophy; leg length discrepancy; trunk focally thick subcutaneous tissue; scoliosis; sandal gap; proximal toe syndactyly; psychomotor delay	Brain MRI: left hemimegalencephaly; thick corpus callosum	*PIK3CA*	c. 3132T > G (p.Asn1044Lys)	21%/nd
**28**	19/M	MCAP	Trunk, limb, and face partly reticulate CM; limb prominent veins; macrocephaly; left hemihypertrophy with macrodactyly; trunk focal thick subcutaneous tissue; sandal gap; proximal toe syndactyly; scoliosis; epidermal nevus; psychomotor delay	Brain MRI: left hemimegalencephaly; thick corpus callosum; Chiari malformation type I; hydrocephalus; cervical syringomyelia	*PIK3CA*	c.241G > A (p.Glu81Lys)	3%/nd
**29**	3/F	MCAP	Trunk and face *** partly reticulate CM; macrocephaly; left lower limb hypertrophy; sandal gap; toe syndactyly	Brain MRI: left hemimegalencephaly	*PIK3CA*	c.2176G > A (p.Glu726Lys)	16%/nd
**30**	4/F	MCAP	Centrofacial ***, trunk, and limb partly reticulate CM; macrocephaly; lower limb overgrowth; toe syndactyly; joint hypermobility; blaschkoid hypochromic macules on the thighs; psychomotor delay	Brain MRI: megalencephaly; polymicrogyria	*PIK3CA*	c.344G > C (p.Arg115Pro)	15%/12%
**31**	6/F	CLOVES	Left flank combined capillary–lymphatic–venous malformation; abdominal phlebectasia; dorsal lipomas; lower limb and left buttock overgrowth; upper limb and shoulder girdle hypotrophy; scoliosis; left foot hexadactyly; toe macrodactyly; widened web spaces; urogenital malformations; psychomotor delay	MRI: trunk and buttock lipomatous overgrowth; flank subcutaneous combined vascular malformation extending to the pelvis and retroperitoneum; terminal filum lipoma	*PIK3CA*	c.3140 A > G (p. His1047Arg)	16.1%/nd
**32**	1/M	CLOVES	Dorsal, abdominal, and flank CM and prominent veins; left flank and lumbosacral lipomas; left lower limb and buttock overgrowth; proximal toe syndactyly; sandal gap; triangular feet	DU: back lipomatous overgrowth	*PIK3CA*	c.3073A > G (p.Thr1025Ala)	5%/nd
**33**	7/M	CLOVES	Right upper limb CM, phlebectasia and lipomatous overgrowth; right thorax combined capillary–lymphatic–venous malformation; right hand hexadactyly and finger syndactyly; trunk lipomatous overgrowth	MRI: right hand lipomatous overgrowth; right upper limb venous malformation; thoracic and abdominal venous–lymphatic malformation; CT scan: multiple lung nodules	*PIK3CA*	c.3140A > G (p.His1047Arg)	46.5%/nd
**34**	10/F	KTS	Left lower limb and abdominal combined capillary–venous–lymphatic malformation; left lower limb overgrowth and pain; leg length discrepancy	DU, CT, and phlebography: abdominal and pelvic phlebectasias; left lower limb deep vein partial agenesis and embryonal superficial veins; portosystemic shunt through umbilical vein remnant	*PIK3CA*	c.325_327delGAA (p.Glu109del)	2.7%/nd
**35**	3/F	KTS	Left buttock and lower limb (including toes) combined capillary–venous–lymphatic malformation; left buttock and lower limb overgrowth and pain; leg length discrepancy	DU and CT: pelvic and left lower limb vein ectasia with persistent embryonic superficial veins; left leg deep vein agenesis	*PIK3CA*	c.3140A > T (p.His1047Leu)	6%/nd
**36**	1/M	Combined CLM	Right flank CM with overlying vesicles	DU: no deep anomalies	*PIK3CA*	c.1633G > A (p.Glu545Lys)	3%/nd ****
**37**	5/M	Combined CVM with overgrowth	Right flank CM with prominent veins and mild overgrowth; proximal toe syndactyly	DU and MRI: abdominal vein ectasia; no AVF	*PIK3CA*	c.325_327delGAA (p.Glu109del)	4%/nd
**38**	6/M	Microcystic LM	Grouped vesicles on intergluteal and right popliteal folds; gluteal swelling	MRI: subcutaneous and intramuscular microcystic lymphatic malformation extending from the buttocks to the right lower limb	*PIK3CA*	c.3140A > G (p.His1047Arg)	2%/nd
**39**	2/M	Combined VLM with overgrowth	Left upper limb normal-colored papules and nodules, bluish vesicles; prominent veins; left upper limb overgrowth	DU: left upper limb vein ectasia with thrombosis; lymphatic microcysts; thickened subcutaneous tissue	*PIK3CA*	c.1633G > A (p.Glu545Lys)	1%/nd ****
**40**	42/F	BRBNS	Diffuse (>100) bluish papules and nodules also on palmoplantar surfaces; violaceous congenital sacral plaque; recurrent gastrointestinal bleeding	MRI: cerebral, hepatic, and osseous vascular nodules; Video capsule endoscopy: numerous gastrointestinal vascular nodules	*TEK*	c.2690A > G (p.Tyr897Cys); **c.2800T > C (p.Ser934Pro)**	1%/nd ****
**41**	9/M	BRBNS	A few bluish papules of the limbs and right plant; violaceous congenital scalp plaque; prominent ipsilateral facial veins	Video capsule endoscopy: a few jejunal and ileal vascular nodules	*TEK*	c.2690A > T (p.Tyr897Phe); **c.2743C > A** **(p.Arg915Ser)**	1%/nd
**42**	11/M	PWS	Left upper limb and thorax reddish patch with pale halo, warmth to palpation; limb hypertrophy and prominent veins; several red-brownish macules on trunk, neck, and upper limbs	DU: left upper limb enlarged arteries with high flow, humeral vein arterialization; MRI: humeral and subclavian vein ectasia, early vein enhancement	*RASA1*	**c.1570_1571insTA (p.Cys525fs*19)**	3%/nd
**43**	12/F	PWS	Left lower limb brownish patches, warmth to palpation; left lower limb overgrowth; pain; leg length discrepancy; trunk macules	DU: arteriolovenulous shunts; vein ectasia without arterialization MRI: vein ectasia	*KRAS*	c.183A > T (p.Gln61His)	4%/nd

AVF = arteriovenous fistula; BRBNS = blue rubber bleb nevus syndrome; CM = capillary malformation; CLM = capillary–lymphatic malformation; CLOVES = congenital lipomatous overgrowth, vascular malformations, epidermal nevi, scoliosis/skeletal and spinal syndrome; CMO = capillary malformation with overgrowth; CT = computed tomography; CVM = capillary–venous malformation; DCMO = diffuse capillary malformation with overgrowth; DU = Doppler ultrasound; KTS = Klippel–Trenaunay syndrome; LM = lymphatic malformation; MCAP = megalencephaly–capillary malformation–polymicrogyria syndrome; MRI = magnetic resonance imaging; nd = not detectable; PWS = Parkes Weber syndrome; SWS = Sturge–Weber syndrome; VAF = variant allele frequency; VLM = venous–lymphatic malformation. * Previously undescribed variants are in bold. ** Affected tissue: skin in all cases, except n. 18, in whom DNA was also obtained from adipose tissue (VAF: 17%), and n. 26, who had a buccal swab. *** Philtrum CM; the patient also carries a germline 15q11.2 microdeletion: arr[GRCh37] 15q11.2(22652330_23272733)x1. **** The known *PIK3CA* mutation c.1633G > A was also detected by droplet digital PCR (ddPCR) with VAF of 3.1% and 0.83% in genomic DNA from lesional tissue in patients n. 36 and 39, respectively (Appendix A). In a similar way, the presence of the previously undescribed c.2800T > C *TEK* variant was confirmed by ddPCR, with a VAF of 1.5% in lesional tissue from patient n. 40 (Appendix A). ddPCR analysis also demonstrated the absence of the abovementioned variants in genomic DNA from peripheral blood of the corresponding patients (Appendix A).

## Data Availability

The data presented in this study are available on request from the corresponding author. The data are not publicly available due to privacy reasons.

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
