# Peer review of "Clinical and Molecular Spectrum of Sporadic Vascular Malformations: A Single-Center Study"

_biomedicines, 2022, doi:10.3390/biomedicines10061460_

Round 1

Reviewer 1 Report

This is generally a well-written paper summarizing well-characterized clinical and molecular features of a group of patients from a single center. Although the findings are not entirely novel, they contribute to an expanding spectrum of such vascular anomalies and genotype-phenotype correlations. 

One main issue/point of confusion lies with the term "sporadic vascular malformation", which suggests a very broad spectrum of malformations that would include very localized lesions, for example arteriovenous malformations, fibroadipose vascular anomalies, intramuscular fast flow anomalies, isolated venous malformations etc. It could, of course, be that the accumulation of cases presented in the study are entirely by chance. However, they seem to represent a pretty specific range on the spectrum of "sporadic vascular malformations", enriched for cases that are more extensive in their presentation. I wonder if the range could be better characterized and defined accordingly. For example, although they are all "sporadic" in the sense of being somatic rather than germline mutations, the vast majority of the cases are syndromic; several involve overgrowth; many are at least segmental. There are a few that are very localized, but the composition does not seem completely representative of the full spectrum of "sporadic vascular malformations". Perhaps the reason for these demographics could be addressed or the term encompassing the cases in this study could be more specific.

Similarly, the statement that "the majority of VMs are rare, chronic and frequently disabling complex disorders" doesn't seem to be true. Many VMs are isolated and don't cause too much trouble -- certainly a subset of VMs, particularly those that are very extensive or are inconvenient locations and particularly the ones that are the focus of this study, are debilitating. But that does not represent the entire breadth of "sporadic vascular malformations": many of them are isolated/localized and possibly asymptomatic.

Were novel/previously undescribed variants confirmed by an orthogonal method?

The manuscript should be proofread again for missing words etc. e.g. "two presented (with) overgrowth" in line 179, but the paper is generally written well.

Author Response

Reviewer 1

Comment 1: One main issue/point of confusion lies with the term "sporadic vascular malformation", which suggests a very broad spectrum of malformations that would include very localized lesions, for example arteriovenous malformations, fibroadipose vascular anomalies, intramuscular fast flow anomalies, isolated venous malformations etc. It could, of course, be that the accumulation of cases presented in the study are entirely by chance. However, they seem to represent a pretty specific range on the spectrum of "sporadic vascular malformations", enriched for cases that are more extensive in their presentation. I wonder if the range could be better characterized and defined accordingly. For example, although they are all "sporadic" in the sense of being somatic rather than germline mutations, the vast majority of the cases are syndromic; several involve overgrowth; many are at least segmental. There are a few that are very localized, but the composition does not seem completely representative of the full spectrum of "sporadic vascular malformations". Perhaps the reason for these demographics could be addressed or the term encompassing the cases in this study could be more specific.

Answer 1: We thank the Reviewer for his/her pertinent comment. Indeed, in our Centre for Vascular Anomalies we routinely perform genetic testing on genomic DNA from lesional tissue selectively in patients with a clinical diagnosis or suspicion of a syndromic vascular malformation or with a segmental vascular malformation involving deep structures. We have now specified the criteria for genetic testing we use in the Materials and Methods section, Study Design paragraph, lines 87-89.

Comment 2: Similarly, the statement that "the majority of VMs are rare, chronic and frequently disabling complex disorders" doesn't seem to be true. Many VMs are isolated and don't cause too much trouble -- certainly a subset of VMs, particularly those that are very extensive or are inconvenient locations and particularly the ones that are the focus of this study, are debilitating. But that does not represent the entire breadth of "sporadic vascular malformations": many of them are isolated/localized and possibly asymptomatic.

Answer 2: We completely agree with this comment and we have modified the sentence which now reads “VMs encompass a wide range of manifestations, from localized isolated skin lesions to highly disabling complex disorders (Introduction section, page 2, lines 50-52).

Comment 3: Were novel/previously undescribed variants confirmed by an orthogonal method?

Answer 3:  Previously undescribed variants were confirmed by repeating targeted sequencing (Materials and Methods section, Molecular genetic testing paragraph, page 3, lines 113-114). In addition, in all cases we performed parallel tissue and blood analysis the latter serving as an internal control for variant confirmation. This has now been better specified in the Materials and Methods section, Molecular genetic testing paragraph, page 3, lines 114-116. Indeed, none of the novel sequence variants identified was detected in blood.

To address Reviewer’s comment, we have now also performed digital droplet PCR (ddPCR) as an orthogonal method. Due to high ddPCR costs, we chose one of the three novel variants, specifically c.2800T>C (p.Ser934Pro) detected at a VAF of 1% by NGS in TEK gene in patient n. 40 affected with blue rubber bleb nevus syndrome (BRBN). The mutation was confirmed with a VAF of 1.5% in genomic DNA from affected skin and was absent in patient’s blood. We have included the ddPCR method in the Materials and Methods section, Molecular genetic testing paragraph, page 3, lines 120-122, the results are mentioned in Table 1 footnote and shown in Supplementary Materials, Figure S1B, together with a detailed method description.  A second previously undescribed variant in TEK gene, c.2743C>A (p.Arg915Ser), was considered likely pathogenic because mutations at codon 915 resulting in different amino acid changes have previously been identified in blue rubber bleb nevus syndrome (BRBN) patients. In addition, the variant was present on the same reads as the known variant c.2690A>G (Supplementary materials, Figure S2) in keeping with the known finding of TEK double mutation in cis in BRBN. The last novel mutation, c.1570_1571insTA (p.Cys525fs*19) in RASA1 is a frameshift that causes a downstream PTC and was thus considered pathogenic.

Please note that, to address Reviewer’s #2 comments, we also performed ddPCR to validate a PIK3CA hot spot variant (c.1633G>A, p.Glu545Lys) detected at very low VAF (3 and 1 %, respectively) by NGS in 2 patients (n. 36 and 39 of Table 1). The findings are reported in Table 1 footnote and shown in Supplementary Materials, Figure S2A.  

Comment 4: The manuscript should be proofread again for missing words etc. e.g. "two presented (with) overgrowth" in line 179, but the paper is generally written well.

Answer 4: We have corrected the phrase and checked the manuscript for additional inaccuracies.

Reviewer 2 Report

In this study, authors identified somatic mutations in patients with different types of sporadic vascular malformations in the genes that are involved in the RAS/MAPK/ERK 16 and PI3K/AKT/mTOR pathways. Some of the mutations in these genes have been reported to lead to vascular malformations before. This is an interesting study and however, the main concern is how solid conclusion that very low variant allele frequencies of variants can lead to VMs.

Main concerns:

  1. Somatic mutations in PIK3CA and other genes have been reported to cause sporadic vascular malformations. A few of patients have low levels of mosaicism less than 5% in tissues. However, in this study, majority of somatic mutations identified have VAF ≤ 5% and more than one-third of mutations have VAF ≤ 3%. Many of the mutations are recurrent mutations. It rises question whether there are a systemic sequencing error or contamination that lead to false positive sequencing results in some of these patients. Therefore, authors need to validate these mutations, especially those that have very low VAF, using a different technique such as droplet digital PCR.
  2. Low VAF of mutations suggests that only very small proportion of cells carry mutations is able to cause vascular and malformations. Authors can perform pathological study to review the pathological changes in tissues, which may provide evidence that small proportion of mutant cells are able to cause diseases.
  3. In the discussion, authors need to discuss the possibility why small proportion of mutant cells is able to spread in different tissues and cause clinical features in patients.

Minor concern:

For the mutations identified, authors can add a table to descript the minor allele frequencies of them in population and bioinformatics prediction of these variants to the protein function.

Author Response

Reviewer 2

Comment 1: Somatic mutations in PIK3CA and other genes have been reported to cause sporadic vascular malformations. A few of patients have low levels of mosaicism less than 5% in tissues. However, in this study, majority of somatic mutations identified have VAF ≤ 5% and more than one-third of mutations have VAF ≤ 3%. Many of the mutations are recurrent mutations. It rises question whether there are a systemic sequencing error or contamination that lead to false positive sequencing results in some of these patients. Therefore, authors need to validate these mutations, especially those that have very low VAF, using a different technique such as droplet digital PCR.

Answer 1: We understand the concern of the Reviewer. However, low levels of mosaicism are typically found in several types of sporadic vascular malformations; for instance the recurrent GNAQ p.Arg183Gln mutation has been detected at VAF <5% in skin biopsies from the majority of cases of capillary malformations and of Sturge-Weber syndrome (Shirley et al. N Engl J Med 2013, 368, 1971-1979: VAF ranging from 1.0 to 18.1% in skin biopsies with a VAF lower than 5% in 12 out of 22 mutated cases; Frigerio et al. PLoS One 2015, 10, e0133158: VAF ranging from 0.85% to 7.42% with VAF less than 5% in 6 out of 9 positive cases; Jordan et al. J Invest Dermatol 2020, 140, 1106-1110: VAF ranging from 1% to 17% with VAF <5% in 8 of 20 GNAQ-mutated cases). The same holds true for GNA11 somatic mutations in diffuse capillary malformation with overgrowth (Couto et al. Angiogenesis 2017, 20, 303-306: VAF ≤ 5% in all cases). As to PIK3CA somatic mutations in sporadic vascular malformations, a recent case series reported that 60% of patients had ≤5% VAF, with a VAF varying from 0.54% to 25.33% (Brouillard et al. Orphanet J Rare Dis 2021, 16, 267). However, this and other studies also found a significantly lower VAF in common and combined lymphatic malformations (median 3.50% in the Brouillard’s study) as compared to disorders of the PROS spectrum (median 8.78% in Brouillard’s study); these findings are in line with ours (PIK3CA mutation VAF 16.1%, 5% and 46.5% in CLOVES lesional skin; PIK3CA mutation VAF range 3-21%, mean 14.7% in the 6 megalencephaly-capillary malformation-polymicrogyria syndrome, MCAP, cases versus 2%, 3%, and 1% in the single cases of microcystic lymphatic malformation, capillary-lymphatic malformation and venous-lymphatic malformation, respectively). Overall, our results appear in keeping with literature findings. Brief comments on this topic have been added in the Discussion section, page 16, lines 383-385, and line 422.

As to the possibility of sequencing artefacts or contaminations, it should be noted that we performed in all cases parallel NGS analysis of patient’s genomic DNA from lesional tissue and from peripheral blood. This has now been better specified in the Materials and Methods section, Molecular genetic testing paragraph, page 3, lines 114-116. Indeed, the sequence variants with a VAF <5% in genomic DNA from lesional tissue were not detected in blood in all but one case, thus strongly arguing against sequencing artefacts and contaminations. Nevertheless, we have now carried out digital droplet PCR (ddPCR) to confirm a PIK3CA hot spot variant (c.1633G>A, p.Glu545Lys) detected at very low VAF (3 and 1 %, respectively) in 2 patients (n. 36 and 39 of Table 1). The mutation was confirmed by ddPCR with a VAF of 3.1% and 0.83%, respectively in genomic DNA from lesional tissue and was absent in blood. In addition, we validated by ddPCR the previously undescribed variant c.2800T>C (p.Ser934Pro) in TEK gene detected at a VAF of 1% by NGS in patient n. 40 affected with blue rubber bleb nevus syndrome. By ddPCR, the mutation was detected at a VAF of 1.5% in genomic DNA from lesional skin and was absent in patient’s blood. We have included the ddPCR method in the Materials and Methods section, Molecular genetic testing paragraph, page 3, lines 120-122, the results are mentioned in Table 1 footnote and shown in Supplementary Material, Figure S1, together with method description. ddPCR testing was not performed in additional cases due to the costs of these assays.    

Comment 2: Low VAF of mutations suggests that only very small proportion of cells carry mutations is able to cause vascular and malformations. Authors can perform pathological study to review the pathological changes in tissues, which may provide evidence that small proportion of mutant cells are able to cause diseases.

Answer 2: This is an intriguing comment and still debated topic. First, it should be considered that we extracted genomic DNA from whole skin biopsies where endothelial cells, which are the main cell type enriched for pathogenetic GNAQ mutations in non-syndromic and syndromic CM (Couto et al. Plast Reconstr Surg  2016, 137, 77e-82e), represent a minor cell population. Moreover, in the case of patient n. 18 affected with diffuse capillary malformation with overgrowth (DCMO) (Table 1) in whom also affected adipose tissue was obtained during surgery, the PIK3CA mutation (p.Glu453Lys) VAF in adipose tissue (17%, Table 1 footnote) was much higher than in a skin biopsy (2%), consistent with what has been previously reported for CLOVES (Kurek et al. Am J Hum Genet 2012, 90, 1108-1115). Indeed, PIK3CA mutations have been detected in different cell types ranging from fibroblasts to adipocytes and endothelial cells (Madsen et al. Trends Mol Med 2018, 24, 856-870). However, variant allele frequencies almost never reach the 50% which would be expected if all tissue cells were carrying the mutation at the heterozygous state. Thus, it has been speculated that PIK3CA mutated cells might exert effects on non-mutated cells by cell-cell interaction or by a paracrine mechanism (Madsen et al. Trends Mol Med 2018, 24, 856-870). A short comment about the different VAF of the PIK3CA mutation p.Glu453Lys in affected skin versus adipose tissue of patient n. 18 has been added in the Discussion section, page 16, lines 424-427.

As to the role of mutant cells in causing the disease, basic studies have indeed shown that GNAQ p.Arg183Gln gain of function mutation drives formation of enlarged blood vessels and have confirmed  the activation of downstream pathways in affected human skin (Huang L et al. Arterioscler Thromb Vasc Biol 2022, 42, e27-e43). As to PIK3CA, various studies in animal models have shown the pathogenetic role of PIK3CA activating mutations in vascular malformations and brain abnormalities (Delestre F et al. Sci Transl Med 2021, 13, eabg0809; for review Madsen et al. Trends Mol Med 2018, 24, 856-870).

As to the request to perform additional studies of pathological changes in tissues, in our paediatric centre we perform small lesional skin biopsies that are entirely employed for DNA extraction. We do not routinely obtain biopsies for histopathological analysis in sporadic vascular malformations. Thus, getting additional biopsies for research purposes would require a novel Ethical Committee approval. It would be beyond the scope of the present study.

Comment 3: In the discussion, authors need to discuss the possibility why small proportion of mutant cells is able to spread in different tissues and cause clinical features in patients.

Answer 3:

The phenotypic variability of patients affected with mosaic vascular anomalies depends on several variables which are only partly known. The most important factor is considered the timing of the mutation, since very early events during embryogenesis may affect a progenitor cell with multiple embryonic destinies causing different lesions in different tissues.  Sturge–Weber syndrome is exemplificative of this theory: GNAQ mutations occurring in the prosencephalon may cause vessels abnormalities in the brain, choroid and forehead skin, while non-syndromic capillary malformations may be due to a late somatic GNAQ mutation. Moreover, the pattern of gene expression will contribute to determine which organ will be affected and the disease behaviour. A further relevant factor in determining disease clinical features is the type of mutation and its functional consequences, as shown for PIK3CA mutations. Finally, as mentioned above, the possibility of paracrine effects should be considered. We have included brief comments on these aspects in the Discussion section, page 16, lines 385-390 and 423-424. 

Comment 4: For the mutations identified, authors can add a table to descript the minor allele frequencies of them in population and bioinformatics prediction of these variants to the protein function.

Answer 4: A Table reporting minor allele frequencies in gnomAD database and pertinent references with bioinformatics predictions and/or functional studies for mutations identified in our series has been added (Supplementary Materials, Table S2) and reference to it made in the text.